# The Process and Challenges of Resident-Led Reconstruction in a Mountain Community Damaged by the Northern Kyushu Torrential Rain Disaster: A Case Study of the Hiraenoki Community, Asakura City, Fukuoka Prefecture, Japan

Yoshio Harada [1], Ai Ichinose [1], Tatsuya Owake [2], Kazuo Asahiro [3], Noriko Sato [4],* and Takahiro Fujiwara [4],*

1 Graduate School of Bioresource and Bioenvironmental Sciences, Kyushu University, Fukuoka 819-0395, Japan
2 Institute for Creative Cities and Regions, University of Hyogo, Kobe 651-2197, Japan
3 Faculty of Design, Kyushu University, Fukuoka 819-0395, Japan
4 Faculty of Agriculture, Kyushu University, Fukuoka 819-0395, Japan
* Correspondence: sato.noriko.842@m.kyushu-u.ac.jp (N.S.); fujiwara.takahiro.218@m.kyushu-u.ac.jp (T.F.)

**Abstract:** Frequent torrential rainfall disasters have occurred worldwide in recent years. In Japan, the Northern Kyushu Torrential Rainfall disaster in July 2017 caused extensive damage to Fukuoka and Oita prefectures and significantly impacted local landscapes, from which residents derive pride and identity and, hence, are of the utmost importance. Local communities in Japan are also at risk of extinction due to progressive depopulation. This study discusses community revitalization through landscape creation and related challenges based on the case of the Hiraenoki Community affected by the Northern Kyushu Torrential Rainfall disaster. We conducted long-term participant observations over two years and semi-structured interviews with all households in the community. We found that the landscape project revitalized local pride and involved numerous people outside the community, including the prefectural extension center, university, and Non-Profit Organization (NPO), and provided them with an opportunity to connect. On the other hand, our investigation also revealed the danger of obscuring the original purpose of reconstruction activities because of collaboration with outsiders. This case study elucidates the possibilities and challenges of resident-led reconstruction activities in communities that are facing depopulation and aging problems and are working with organizations within and outside the community.

**Keywords:** local landscape; observation; depopulation; municipal government; local identity; revitalization; outsiders; collaboration; resilience

## 1. Introduction

In recent years, floods and landslides caused by torrential rainfall have occurred frequently worldwide. In Japan, the Northern Kyushu Torrential Rainfall disaster in July 2017 (NKTR), torrential rainfall in July 2018, the East Japan Typhoon in October 2019, and torrential rainfall in July 2020 caused extensive damage to many areas. These are enumerated in the list of meteorological, seismic, and volcanic phenomena named by the Japan Meteorological Agency [1]. According to the Japan Meteorological Agency, the annual number of occurrences of ≥80 mm of precipitation per hour and the annual number of days with ≥400 mm of precipitation per day increased significantly between 1976 and 2020 [2]. Therefore, the frequency of cases of intense rainfall over a short period is increasing in Japan.

The NKTR was caused by a linear precipitation system that was formed and sustained by the effect of warm and very humid winds flowing into a stationary seasonal rain front in the vicinity of the Tsushima Strait during 5–6 July 2017 [3]. As a result, continued torrential rain occurred, resulting in record heavy rain in northern Kyushu, such as Asakura City and Toho Village in Fukuoka Prefecture and Hita City in Oita Prefecture [3]. The total

precipitation for 2 days in northern Kyushu peaked at >500 mm in certain areas, and new observation records (24-h precipitation) were recorded in Asakura City (545.5 mm) and Hita City (370.0 mm) [3]. This record-breaking precipitation caused severe damage in Fukuoka and Oita prefectures, including 40 casualties and 2 missing persons, and more than 1600 houses were completely or partially destroyed and inundated above the floor level [3]. In addition, torrential rain has severely damaged utilities, such as water supplies and electricity, as well as roads, railways, agriculture, and forestry, which are key industries in this region [3]. Slope failures and mudslides are frequent in mountainous areas and cause large amounts of driftwood [3]. As a result of the NKTR disaster, the Japanese government designated Asakura City, Toho Village, Soeda Town in Fukuoka Prefecture, and Hita City in Oita Prefecture as having experienced "severe disaster areas" following the related law.

Disasters caused by torrential rain also significantly impact the local landscape owing to landslides, housing damage, and the associated reconstruction work. The Landscape Law of Japan (promulgated in 2004) states that "good landscapes are indispensable for creating a beautiful and dignified land and an enriched living environment" (Article 2, Clause 1) [4]. According to this law, "good landscapes are formed through harmony between nature, history, culture, etc. and people's daily lives, economic activities, etc. in each region" (Article 2, Clause 2) and "good landscapes are closely related to the unique characteristics of each region" (Article 2, Clause 3) [4]. Consequently, the local character and regional identity of mountain village communities may be affected by damage to local forest landscapes caused by torrential rainfall disasters. Rehabilitating local landscapes in disaster-affected areas is necessary but not sufficient to recreate beautiful landscapes. Rather, rehabilitation should serve as a disaster prevention and mitigation system utilizing ecosystem services while considering the characteristics of the unique industries that constitute the identity of the region.

Local landscapes are essential to residents who derive pride and identity from them. Therefore, efforts aimed at increasing the willingness of victims to reside in the disaster-affected areas and helping their villages survive through landscape restoration can be considered "creative reconstruction" [5]. Currently, Japan is facing concerns regarding increasing torrential rainfall disasters destroying local landscapes. There have been reports of local governments being unable to adequately respond to disaster recovery owing to the downsizing of administrative operations, including personnel cutbacks [6]. Conversely, local communities have previously played the roles of assisting with dealing with matters that cannot be handled by individuals or families alone, maintenance of local culture, general interest coordination, liaison and coordination between the government and residents, and supplemental functions for the government [7]. Therefore, in situations with limited local government capacity, the role of local communities in the disaster recovery process is likely to be significant. The Reconstruction Design Council in Response to the Great East Japan Earthquake also states that "Given the vastness and diversity of the disaster region, we shall make community-focused reconstruction the foundation of efforts towards recovery. The national government should support reconstruction through general guidelines and institutional design" in the Seven Principles for the Reconstruction Framework [8]. Therefore, the Japanese government is emphasizing the role of local communities in disaster recovery.

Certain recovery activities that do not rely on local governments were reported. For example, in the case of the 2016 Kumamoto earthquake, reconstruction activities centered on community development councils were undertaken in rural villages, which account for most of the affected area [9]. In the case of the NKTR, an agricultural cooperative collaborates with the Non-Profit Organization (NPO) to accept volunteers and plays a role in connecting stakeholders, thus contributing to the recovery of local agriculture [10]. However, depopulation and aging are serious concerns in Japan owing to the rapid population decline, particularly in rural and mountainous areas; moreover, local communities are at risk of extinction [8]. This trend is even more pronounced in communities where the population has been reduced by natural disasters.

Many studies on disasters and resilience were conducted outside Japan. For example, Eakin et al. investigated the linkages between household vulnerability and resilience through the case of torrential rains associated with Hurricane Stan which devastated farm systems in southern Mexico in 2005 [11]. The authors argued that policy interventions not only enable individual survival but also enhance resilience at local, community, and landscape scales, helping to provide local strategies and knowledge on risk management [11]. In Pakistan, which is considered highly vulnerable to climate change impacts, Memon and Ahmed indicated that the lack of multi-sectoral productive economic opportunities had a negative impact on the resilience of rural households and women, and that female-headed households were more vulnerable than male-headed households [12]. Sun et al. identified the disaster types historically faced by rural settlements in Xinjiang, China, and divided the landscape carrier based on the evolution of these settlements [13]. The authors also proposed the resilience mechanism of adaptation to disasters for rural communities in Xinjiang based on the experience of disaster resilience and adaptation in traditional rural settlements [13].

Some European countries are also facing depopulation problems. MacDonald et al. reported a decline in traditional labor-intensive practices and abandonment of marginal agricultural land in many areas, particularly in mountain areas [14]. Lasanta et al. also mentioned that farmland abandonment had a far greater impact on mountain areas because of rural depopulation as well as biophysical constraints and that it would continue in the following decades [15]. Westhoek et al. carried out a scenario study (termed EURURALIS) to stimulate the strategic discussion among national and European Union policymakers on the future of rural areas in Europe and the role of policy instruments [16].

Therefore, it appears that the case of resident-led reconstruction in a mountain community damaged by a natural disaster in Japan can provide valuable insights and information for further discussions on the resilience of communities affected by natural disasters and depopulation. There are many studies focusing on community recovery processes in Japan; however, the majority are cases of communities affected by earthquakes such as the Great East Japan Earthquake [17,18] and few studies have been conducted on torrential rain disasters. However, the potential destruction of local landscapes by torrential rain disasters in Japan is concerning. Given the limited capacity of local governments, the role of local communities in disaster recovery is regarded as crucial. In addition, local communities in rural areas are at risk of extinction. How is the restoration of local landscapes in rural areas affected by torrential rainfall disasters? How have these landscapes affected the revitalization of local communities? What challenges do these landscapes face? To the best of our knowledge, no previous study has addressed these questions.

Therefore, this study aimed to discuss community revitalization through landscape creation and related challenges in the Hiraenoki Community, Asakura City, Fukuoka Prefecture, an area affected by the NKTR.

## 2. Materials and Methods

### 2.1. Study Site

We conducted field research in the Hiraenoki Community, Asakura City, Fukuoka Prefecture, Japan. The present "Hiraenoki" Community was formed by merging two communities ("Daira" and "Enoki" communities) during the Meiji period. Although the two communities have become less distinct, residents remain aware of the distinction. Currently, 20 households live in the Hiraenoki Community, 1 of which is an immigrant from outside the community. In addition, two households live outside the community as "semi-community residents" and commute to their homes purchased in the Hiraenoki Community.

The Hiraenoki Community is located on a fan-shaped site. The community has no rice paddy fields; rather, most residents cultivate persimmons, the main agricultural crop in the community. Original Shiwa persimmon (a persimmon species in Japan) trees planted in 1926 remain in the community, and residents have been producing persimmons for >100 years [5]. These persimmon orchards are an essential income source for residents. They also form the center of the local landscape and have become a symbol of the community (Figure 1).

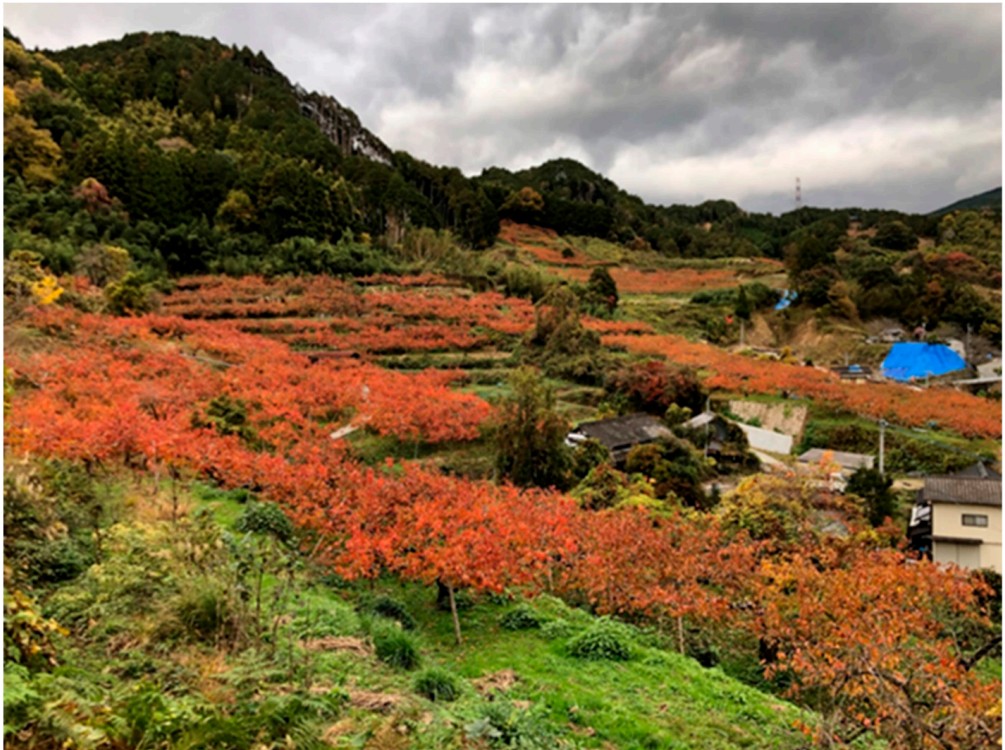

**Figure 1.** Local landscape formed by persimmon orchards in the Hiraenoki Community (obtained by Yoshio Harada [the first author] on 28 November 2019).

The Hiraenoki Community was severely damaged by the NKTR. Although there were no fatalities in the community, some houses were completely or partially destroyed, and the mountain slopes collapsed in numerous areas. When we conducted this study in the community in 2019, i.e., more than two years after the disaster, slope protection construction was ongoing in several places. Roads within the community were severely damaged. The road to the south of the community had been restored, whereas the road to the north had not. Persimmon orchards were also severely affected by the NKTR. For example, landslide damage in certain areas completely inhibited persimmon production. In other cases, roads leading to persimmon orchards were damaged and became impassable, and persimmon orchards were bought to establish facilities for erosion control. As a result of the NKTR, the population decreased from 37 households (83 residents) before the NKTR to 19 households (45 residents) as of January 2020. In addition, some residents felt that the landscape had deteriorated owing to fallen Japanese cedar (*Cryptomeria japonica*) and cypress (*Chamaecyparis obtuse*) trees from landslides in forests, bamboo encroachment, and slope protection construction (Figure 2).

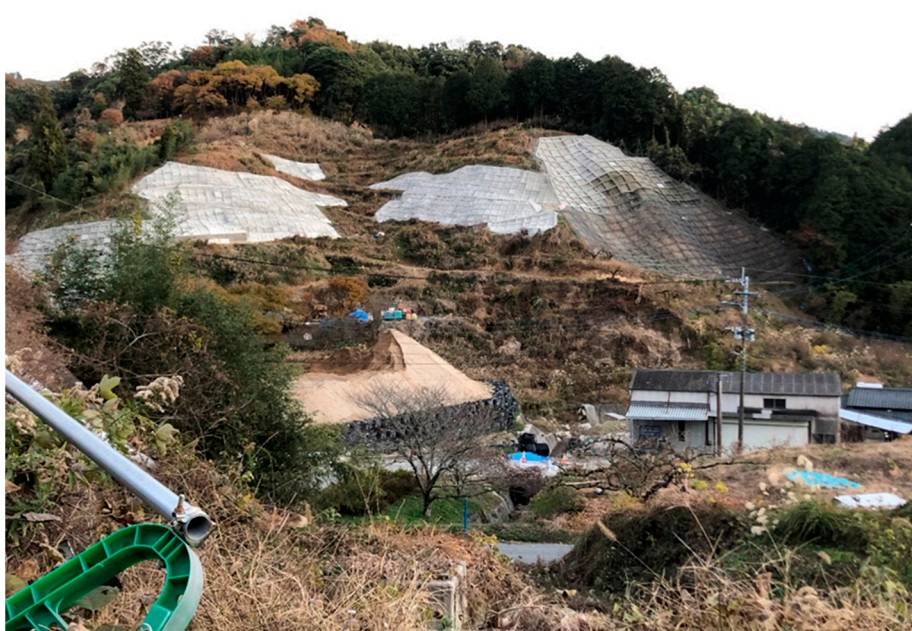

**Figure 2.** Ongoing road restorations and slope protection constructions in the Hiraenoki Community (obtained by Yoshio Harada [the first author] on 12 December 2019).

*2.2. Hiraenoki Reconstruction Committee*

After the NKTR disaster in July 2017, volunteers mainly restored and reconstructed the living environment, including the restoration of infrastructure and houses, as well as the removal of soil and sand encroaching on the roads. Government-led work on collapsed slopes and the restoration of rivers and roads was ongoing at the time of the present study. The Hiraenoki Reconstruction Committee (HRC) was based on the "Road Committee," which was formed by several residents before the NKTR. The Road Committee conducted activities such as reporting collapsed roads in the community to the local government and collecting money for road repairs.

When the living environment for residents in the community was secured to a certain extent through restoration work by volunteers and the government, the HRC was established by agreement among the residents in the regular community meeting on 14 April 2019. The establishment of the HRC was based on the residents' sense of crisis regarding the survival of the community, and the committee members' desire to "make the community a place where residents and others are happy to live."

The HRC comprises seven members, i.e., six residents and a community head, and has three main objectives: (1) community development, including early completion of restoration projects, creation of a living environment, and safety measures; (2) maintenance of persimmon orchards in the community for ≥10 years; and (3) creation of a local landscape. Landscape creation is currently the main activity of the HRC and mainly involves the maintenance of the "Kunugiyama Observation" scenic overlook in the community. The landscape creation is funded by the community budget.

During the establishment of the HRC process, the opinion arose that "it would be difficult for community residents alone to carry out reconstruction activities". Therefore, the HRC chairperson consulted an acquaintance who was an alderman. The alderman then introduced the Asakura Extension and Guidance Center of Fukuoka Prefecture (hereafter, Asakura Extension Center) and the Kyushu University Disaster Recovery and Reconstruction Support Team, including the Faculty of Agriculture and Faculty of Design, Fukuoka (hereafter, Kyshu University Team), and requested assistance with the construction activities. Consequently, the Asakura Extension Center and Kyushu University Team participated in activities commencing in June 2019.

The main activities of the HRC include information-sharing meetings held once a month with the HRC members, Kyushu University Team, and Asakura Extension Center; landscape creation work with the community residents; preparation of grant applications for HRC activities; coordination of community events; and petitions to the local government regarding the infrastructure in the community.

*2.3. Data Collection*

We conducted long-term participant observations between June 2019 and January 2022. We participated in activities organized by the HRC, focusing on local landscape creation, hearing the residents, and observing and recording these activities. All regular meetings by the HRC were in-person meetings. In addition, we conducted semi-structured interviews with all the households in the community in January 2022. Question items included (1) awareness of HRC activities; (2) participation in HRC activities; (3) expectations of the HRC; (4) evaluation of HRC activities; (5) challenges faced by HRC; (6) participation in the Hiraenoki Reconstruction Tree Planting Ceremony; (7) visits to the observation site outside of maintenance activities; (8) impression of the landscape at Kunugiyama observation site; (9) impression of the landscape seen from the Kunugiyama observation site; and (10) interaction inside and outside the community after the establishment of the observation site.

**3. Results**

*3.1. Kunugiyama Observation Site Creation Process*

3.1.1. Before the Determination of the Proposed Observation Site (by February 2020)

The main HRC activity in 2019 was developing a policy for reconstruction activities, including identifying the current status of the community after the NKTR disaster and projections for the future. In addition to the exchange of opinions among HRC members, the attractiveness of the community was confirmed through interactive meetings with the community's women's club and ex-residents who relocated from the community after the NKTR. Consequently, the aesthetics of the autumn foliage of persimmon trees and the taste of the well water were identified as the most attractive features of the community.

Based on the opinions of the residents, certain proposals were made such as organizing the river running through the community into a playground for children, promotion of traditional events in the community, and utilization of the community hall. Finally, the proposal, which revitalizes the community through landscape creation by establishing an observation site overlooking the persimmon orchards in the community, was adopted at the autumn persimmon foliage viewing event on 28 November 2019, to which the Kyushu University Team and the Asakura Extension Center were invited.

The HRC decided to plant broadleaf trees that would add color to the vegetation in the community during autumn. The Kyushu University Team assisted in determining the species and location of planted trees. Initially, there were two candidate sites for observation: one on the west side of the community near Kannondo Temple and one on the east side near the Kunugiyama Mountain. The candidate site was decided upon by voting after the autumn foliage viewing event on 28 November 2019. The western side was initially selected; however, the eastern side was ultimately selected as the observation site in February 2020 because a cemetery was located on the western side.

3.1.2. Concretization of the Observation Site Creation (by the Hiraenoki Reconstruction Tree-Planting Ceremony)

After the observation site was determined in February 2020, HRC meetings were suspended owing to the COVID-19 pandemic but resumed in July 2020. The species and size of the seedlings to be planted in accordance with the site conditions and the committee's requirements were then determined in a series of HRC meetings.

The proposed observation site was originally a persimmon orchard that was no longer manageable because of the NKTR. The HRC signed a contract with the three

landowners to lease land (2183 m$^2$) for 20 years (from 2020–2040). In addition, it was necessary to clear the remaining persimmon trees before planting broadleaf trees. Therefore, community residents, Kyushu University Team members, and Asakura Extension Center staff mowed the grass and cleared the remaining persimmon trees. They also built roads and installed drainage ditches in the planned area. Simultaneously, the Kyushu University Team surveyed and mapped the site.

In September 2020, based on the survey results and budget, the number of trees to be planted and their locations were determined based on the advice of Kyushu University Team members. In addition, a detailed plan for the observation site, created by a faculty member of the Faculty of Design of Kyushu University (the fourth author of this paper), was availed to the public to share the image of the completed project (Figure 3). In October 2020, stakes were driven into the planting areas, and weed prevention sheets, deer nets, and simple toilets were installed.

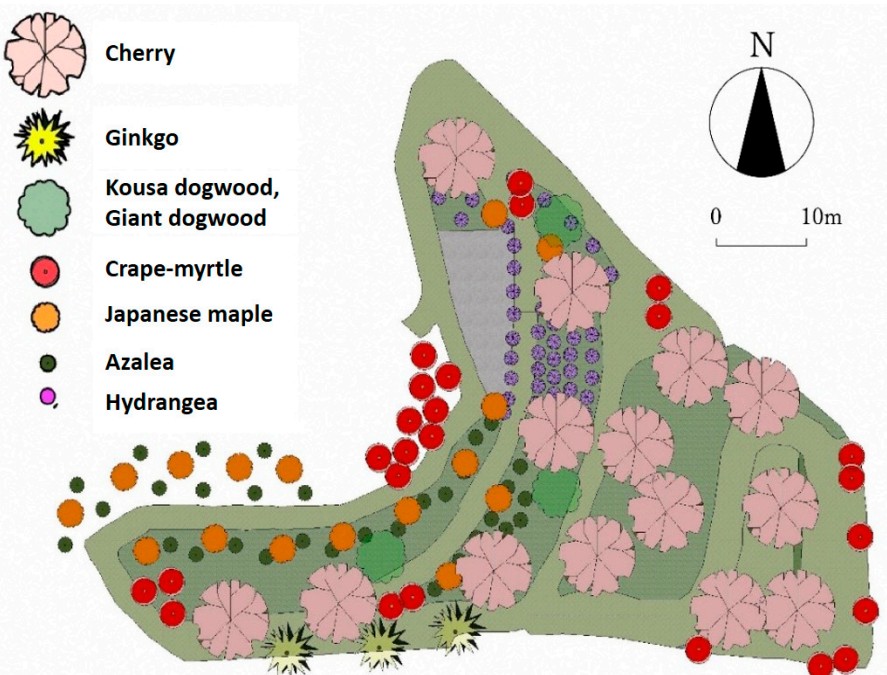

**Figure 3.** Plan of the observation site (created by Kazuo Asahiro [the fourth author]).

At the HRC meeting in November 2020, the Kyushu University Team introduced NPO Asa-Kuru to the HRC, and they decided that the two organizations would collaborate. It was also decided that the tree planting at the observation site would be held as a "Hilaenoki Reconstruction Tree-Planting Ceremony", inviting residents and ex-residents, staff of Asakura Extension Center and Kyushu University Team, children participating in NPO Asa-Kuru activities, and the media to help disseminate information regarding the project. It was also decided that the ceremony would be held to communicate with the children of the Shiwa School District, where the Hiraenoki Community is located.

In December 2020, HRC planted cherry (*Cerasus* Mill.), crape-myrtle (*Lagerstroemia indica*), and Ginkgo *(Ginkgo biloba* L.) trees and confirmed the location of a signboard before the ceremony (Figure 4). In the subsequent meeting in January, detailed arrangements and roles for the ceremony were determined and a pamphlet was prepared for distribution.

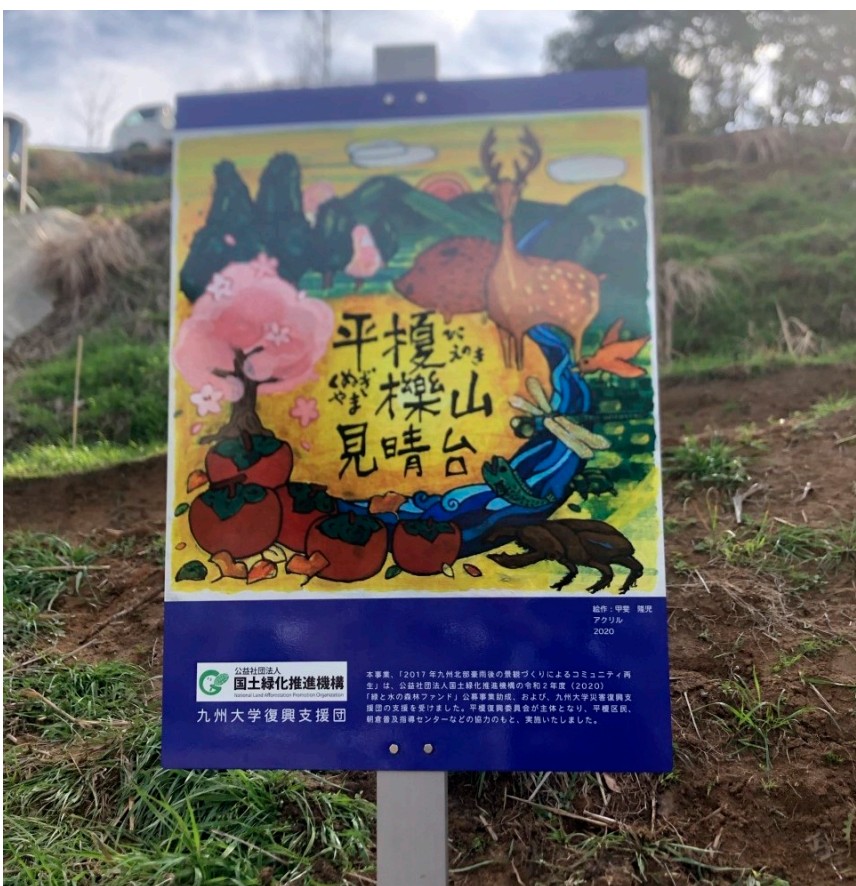

**Figure 4.** Signboard at the Kunugiyama observation site (obtained by Yoshio Harada [the first author] on 6 March 2021). The signboard notes that this project was led by the HRC and the community residents with support from Kyushu University Team, Asakura Extension Center, and the National Land Afforestation Promotion Organization.

3.1.3. Hiraenoki Reconstruction Tree-Planting Ceremony (6 March 2021)

The Hiraenoki Reconstruction Tree Planting Ceremony was held on 6 March 2021. In total, 110 people participated in the event. Regarding the participation of ex-residents, the HRC sent invitations to 14 people, 4 of whom participated.

Children mainly planted small seedlings, such as hydrangeas (*Hydrangea macrophylla*) and azalea (*Rhododendron* L.), and adults planted medium-sized seedlings, such as Japanese maple (*Acer palmatum* Thunb.). In total, 144 trees, consisting of 9 Someiyoshino cherry (*Prunus yedoensis* Matsumura), 4 Yamazakura cherry (*Cerasus jamasakura*), 4 ginkgo, 2 kousa dogwood (*Cornus kousa*), 1 giant dogwood (*Cornus controversa*), 25 crape-myrtle, 20 Japanese maples, 45 hydrangeas, and 34 azaleas were planted. The planted trees were labeled with the name tag of the person who planted the tree, which was intended to strengthen the connection between the participants and the Hiraenoki Community. At the tree-planting ceremony, children played musical instruments (Kalimba) that were handmade from driftwood generated during the NKTR disaster. The ceremony was reported in newspapers. Subsequently, in April 2021, a group photo of the tree-planting ceremony participants and a project report were distributed. In addition, panels were set up to introduce the past and present of the Hiraenoki Community and the Kunugiyama observation site (Figure 5).

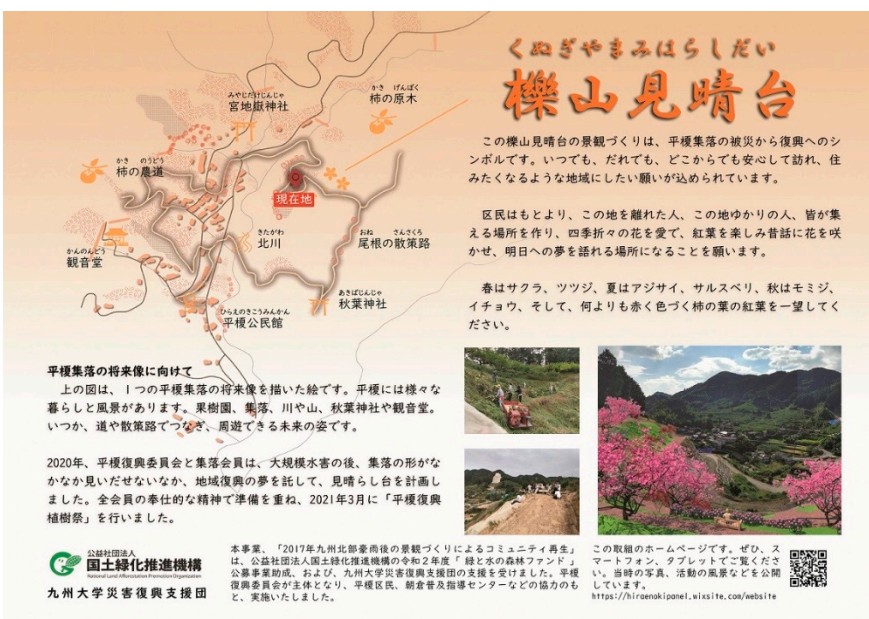

**Figure 5.** Panel installed at Kunugiyama observation site (created by Kazuo Asahiro [the fourth author]). The panel shows the community residents' thoughts on the Kunugiyama observation as a symbol of reconstruction and their vision of the future of the Hiraenoki community. It also includes a link to a website that introduces the Kunugiyama observation project with a message from HRC and community residents. Please access the website on your smartphone or tablet. Photos from those days, activity scenes, etc. are available.

The observation site project was subsidized (800,000 yen) by the Green and Water Forest Fund of the National Land Afforestation Promotion Organization, and the committee allocated approximately 500,000 yen from their community budget. To minimize costs, all work at the observation site by residents was carried out voluntarily. Consequently, labor costs were reduced by approximately 200,000 yen from the original estimate. In addition, residents utilized their personal connections to obtain saplings at low prices and by division from trees in the community.

During the tree-planting ceremony, we conducted a simple interview with four ex-residents who participated in the tree-planting ceremony. The following three questions were posed: (1) how did you know about the tree-planting ceremony?, (2) do you still own persimmon orchards in the Hiraenoki Community?, and (3) do you want to live in the Hiraenoki Community again?

All four respondents answered that they were informed of the tree-planting ceremony by invitation from the HRC in January 2021. Thus, they were unaware of the Kunugiyama observation site prior to receiving the invitation. In terms of persimmon orchards, three of the four respondents answered that they did not own a persimmon orchard because they had already stopped producing persimmons before the NKTR. One respondent still owned a persimmon orchard in the Hiraenoki Community and visited the community at least once a month to maintain it. When we asked the respondents if they would want to live in the Hiraenoki community again, all four answered "No". Three respondents stated that they could no longer live in the Hiraenoki Community due to the trauma of the disaster experience.

The interview results indicated that it would be very difficult for the community members to reside in the Hiraenoki Community again. On the other hand, we found that their relationship with the Hiraenoki Community had not completely disappeared. For example, an ex-resident visited the Hiraenoki Community to maintain his persimmon orchards. The invitation to the tree-planting ceremony from the HRC also served to connect them with the community.

3.1.4. Activities after the Completion of the Kunugiyama Observation Site (from March 2021)

In June 2021, the HRC, NPO Asa-Kuru, and Kyushu University Teams formed the "Hiraenoki Community Guard Group (*Hiraenoki Satomori Kai*)". As part of these activities, they decided to invite children participating in Asa-Kuru to the Yodo Festival on 23 July 2021, a Shinto ritual held every other year in the Hiraenoki Community, and also to implement recreational activities for them. The Yodo festival is a ritualistic event with fireworks at the Akiba Shrine. The festival was also an opportunity for residents to communicate with each other before the NKTR. At the 2021 Yodo festival, Hiraenoki Community Guard Group staff made preparations in the morning and the children participated in recreation activities in the afternoon, including nature games prepared by the Kyushu University Team, shaved ice, ball scooping, river games by NPO Asa-Kuru, and fireworks prepared by the HRC. The recreational fireworks display was held in front of the community center (Figure 6). Residents also held a firework display at the Akiba Shrine as a Shinto ritual.

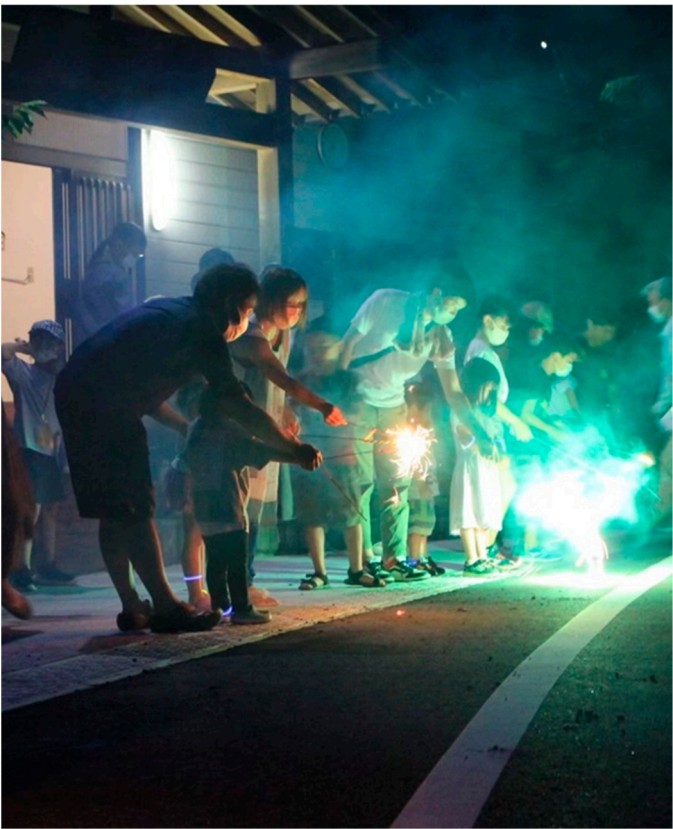

**Figure 6.** Yodo festival in which children participated (obtained by Ai Ichinose [the second author] on 23 July 2021).

Recreational activities with residents and children from the Asa Kuru Community were held at the Reconstruction Autumn Foliage Viewing Event on 28 November 2021 (Figure 7). During the event, a walk around the observation site and a viewing of autumn foliage were held to promote the beauty of autumn foliage in the persimmon garden in the community. The residents served persimmons to the participants and staff from the Asakura Extension Center gave small lectures on persimmon varieties to the children. In addition, a treasure hunt game prepared by the Kyushu University Team and a puppet show by a street performer commissioned by Asa-Kuru were implemented.

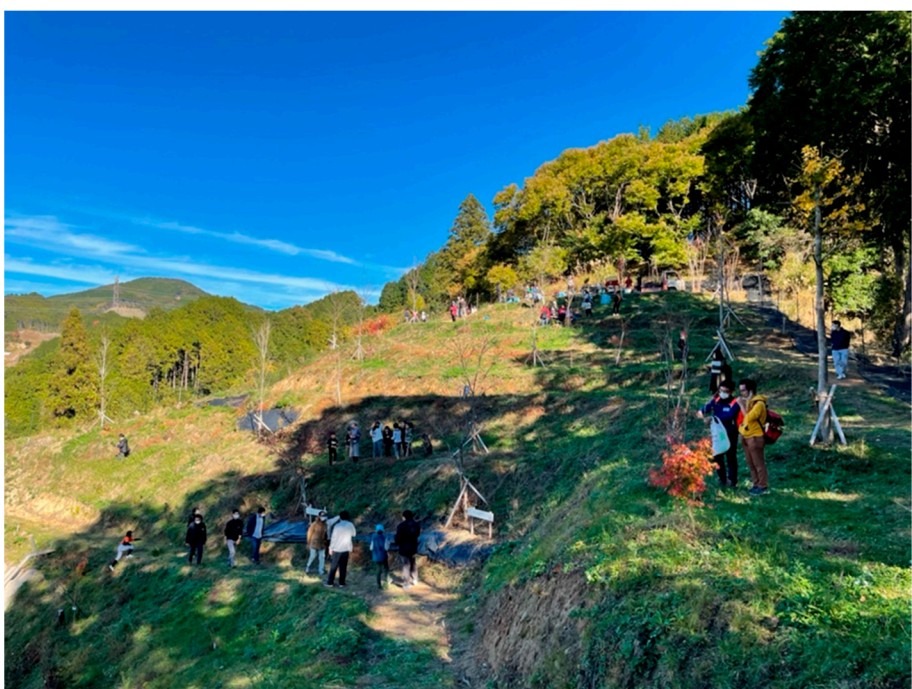

**Figure 7.** Reconstruction Autumn Foliage Viewing Event (obtained by Yoshio Harada [first author] on 28 November 2021).

This event also served as an opportunity to assess the condition of the observation site after the tree-planting ceremony. At the meeting after the event, the participants shared their concerns about the large number of pests that had eaten the leaves of the cherry trees and the fact that the wild boars had broken through the nets. It was also confirmed that four crape-myrtle trees and one Japanese maple tree had died. Therefore, the HRC decided to replant these 5 trees and an additional 16 azaleas and 32 hydrangeas. The replanted tree species are widely found in Japan.

*3.2. Results of Semi-Structured Interviews with Community Households*

The following opinions on the Hiraenoki Reconstruction Committee were recorded in this study.

3.2.1. Awareness of HRC Activities

Of the 18 respondents, 17 were aware of HRC activities (Table 1). However, of the 10 respondents who stated that they "knew" (excluding the committee members), 7 stated that they only knew about the observation site creation activity. The other three respondents were aware of beautification activities in the community, working with students and children, attending information meetings, and lobbying the local government. The one respondent who answered "don't know" stated that "I was not aware of the activities of the reconstruction committee but was aware of the existence of the observation site".

**Table 1.** Resident awareness of the Hiraenoki Reconstruction Committee (HRC) activities.

| Know | Not Know |
| --- | --- |
| 17 | 1 |

3.2.2. Participation in the Hiraenoki Reconstruction Committee Activities

Of the 11 respondents (excluding the committee members), 9 participated every time they were called upon if possible, for example by mowing the grass at the observation site (Table 2).

**Table 2.** Participation in HRC activities by residents (excluding committee members).

| Participated | Did not Participate |
|:---:|:---:|
| 9 | 2 |

In addition, 11 residents (non-HRC members) were asked whether they wanted to participate more in HRC activities (Table 3). All the residents said "no" because they had already fully participated in the activities. Other reasons given by the residents who answered "no" included "age", "I can't do anything if I interfere", "participation does not benefit the residents", and "I feel I was forced to participate".

**Table 3.** Willingness of residents regarding participation in HRC activities (excluding HRC members).

| I Want to Participate More | Status Quo Is Good Enough |
|:---:|:---:|
| 0 | 11 |

In contrast, HRC members were asked if they wanted residents to participate more in committee activities. Three of the seven members answered that they wanted the residents to participate more actively (Table 4). Their reasons were "not enough participation by the population below the early 60 s" and "I feel that the residents' cooperation is still only approximately 80% at the moment". Conversely, the HRC members who answered "I don't think so" had the following opinions: "we can't force them to participate" and "we would like to expand our activity to the outside of the community and cooperate with other communities rather than inside of the community".

**Table 4.** HRC members perspectives regarding resident participation in HRC activities.

| I Want the Residents to Participate More | I Don't Think So |
|:---:|:---:|
| 3 | 4 |

3.2.3. Expectations of the Hiraenoki Reconstruction Committee

The question was asked in an open-ended format. The most common answer was "maintenance and management of the observation site and its further development" (six respondents). Two respondents answered "early completion of restoration work" and "safety measures for the community and observation site". Other responses were as follows: "maintaining the current situation", "releasing fish into the river", "nursery for fireflies", "organizing abandoned farmland", "creating a lively landscape", "installing a walking trail in the community", "creating a place that can accommodate numerous people", "creating a place for visitors to stop by", "not to be second best", "promotion activities for the observation site", "having more children participate", "calling on the administrative construction office to plant trees for erosion control", "holding social gatherings attended by ex-residents", "arranging for flower viewing", "maintaining and utilizing the Kannondo temple and other facilities in the community effectively", "activities that will properly benefit the community", "activities for the internal residents", and "support for residents' marriages".

3.2.4. Evaluation of the Hiraenoki Reconstruction Committee Activities

Regarding whether the HRC activities aligned with their expectations, seven respondents answered "in line", two answered "not in line", and nine answered "can't say either way" (Table 5). The most common reason for "undecided" was that they could not make a judgment because the results were not yet available. Residents who responded "not in

line" felt that the HRC activities were not beneficial to the community and expected the HRC to target the the Hiraenoki Community residents.

**Table 5.** Evaluation of HRC activities.

| In Line | Not in Line | Can't Say Either Way |
|---------|-------------|----------------------|
| 7 | 2 | 9 |

3.2.5. Challenges of the Hiraenoki Reconstruction Committee

The question was asked in an open-ended format. The most common response was a decrease in the number of bearers owing to depopulation and aging, and the resulting difficulties in long-term maintenance and management" (10 respondents). The second most common response was "There is a difference in views and motivations for reconstruction activities between generations, and relatively young residents do not cooperate well" (7 respondents).

Other opinions expressed by several respondents were that "the observation site has not yet become a gathering place" (4 respondents) and "safety measures have not been taken at the observation site and along the path, posing a risk of injury" (3 respondents). There were also a few comments interpreted as dissatisfaction with the HRC, such as, "The HRC is not open to women's participation", "Insufficient explanation to the residents", and "The HRC is closed and only the leader's opinion is strong".

The following were opinions on the Kunugiyama Observation Site.

3.2.6. Participation in the Hiraenoki Reconstruction Tree Planting Ceremony

Eleven respondents said they "participated and planted trees", three said they "participated but did not plant trees", and four said they "did not participate" (Table 6). The three respondents who answered "participated but did not plant trees" indicated that they were not willing to participate because they felt forced to do so and that they wanted to help the event but felt it was irresponsible to plant trees that they had no way of managing. The four respondents who "did not participate" gave the following reasons: "my schedule did not allow me owing to other commitments", "injury", and "age".

**Table 6.** Participation in the Hiraenoki Reconstruction Tree Planting Ceremony.

| Participated and Planted Trees | Participated But Did Not Plant Trees | Did Not Participate |
|--------------------------------|--------------------------------------|---------------------|
| 11 | 3 | 4 |

3.2.7. Visits to the Observation Site Outside of Maintenance Activities

Seven respondents sometimes visited observations outside of the maintenance work. When asked if it was difficult for them to visit the site, eight respondents answered "not hard", seven answered "hard but want to visit", and three answered "hard, so I don't want to visit" (Table 7). This indicated that the majority felt that it was a burden to visit the observation site. Two of the three respondents who answered that they did not want to visit the observation site were relatively young men in their 50s.

**Table 7.** Opinion regarding the difficulty in visiting the observation site.

| Not Hard | Hard, But Want to Visit | Hard, I Don't Want to Visit |
|----------|-------------------------|-----------------------------|
| 8 | 7 | 3 |

### 3.2.8. Impression of the Landscape at the Kunugiyama Observation Site

When asked if the observation site "feels like a typical Hiraenoki landscape", a photograph was presented to the residents so they could answer based on a common understanding of the question (Figure 8). As a result, eight answered "feels like the typical landscape like Hiraenoki", seven answered that it "does not feel like a landscape like Hiraenoki", and three answered "can't say either way" (Table 8).

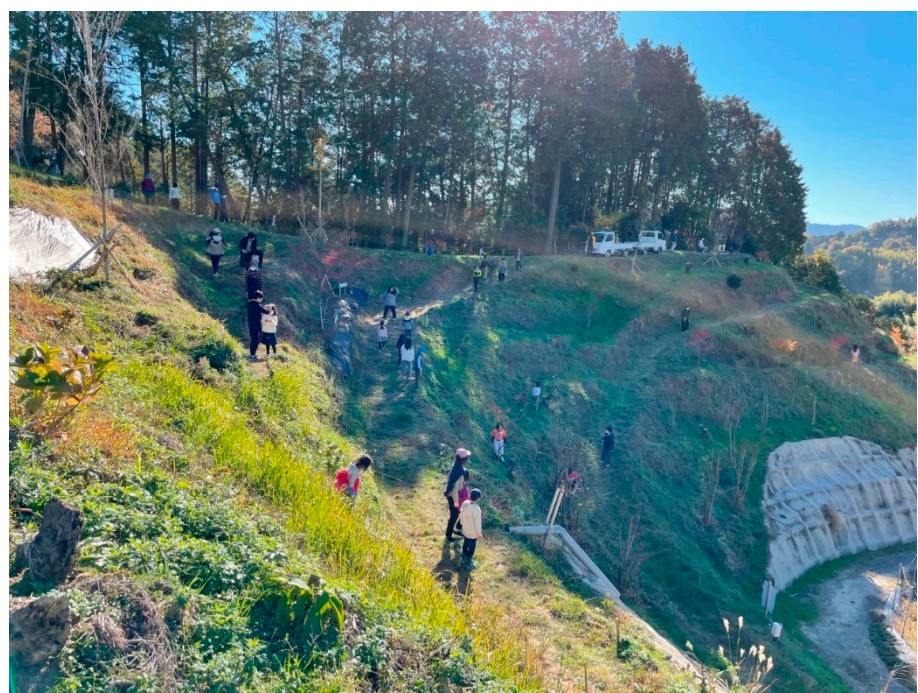

**Figure 8.** Kunugiyama observation site (obtained by Yoshio Harada [the first author] on 28 November 2021).

**Table 8.** Opinion on the Kunugiyama observation site as the local landscape of the community.

| Feels Like the Typical Landscape Like Hiraenoki | Does Not Feel Like Hiraenoki Landscape | Can't Say Either Way |
|---|---|---|
| 8 | 7 | 3 |

The reasons given for "feels like Hiraenoki landscape" were "because it could be a new symbol of Hiraenoki Community" (5 respondents), "because it is a place of relaxation" (2 respondents), "because it is better to have it than not to have it" (2 respondents), "because it was built by everyone working together" (1 respondent), and "because it is my ex-persimmon orchard" (1 respondent). In contrast, the reasons given for "not feeling like Hiraenoki Community landscape" were "It is an unfinished project, and no results have been achieved so far" (2 respondents), "It is not very familiar yet in the community" (1 respondent), "There are no fruit trees, which is the industry of Hiraenoki Community" (1 respondent), "It looks like a landscape that can be found anywhere" (1 respondent), and "The original Hiraenoki landscape is not here" (1 respondent). The reasons given for "can't say either way" were "it's my first time here, and I don't know" and "it's an unprecedented landscape."

### 3.2.9. Impression of the Landscape View from the Kunugiyama Observation Site

The research question "Do you feel that the view from the observation site is the typical landscape of Hiraenoki?" was answered after a photograph was presented to the residents so that they could answer based on a common understanding (Figure 9). Thirteen

responded that the view was "landscape like Hiraenoki," four answered that it was "not landscape like Hiraenoki," and one answered, "can't say either way" (Table 9).

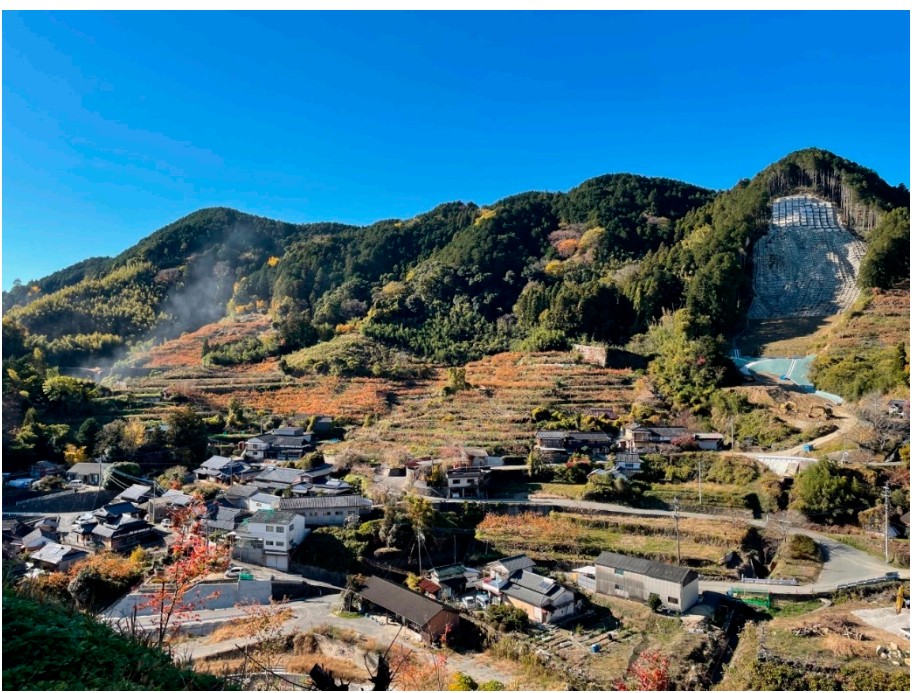

**Figure 9.** Landscape from the Kunugiyama observation site (obtained by Yoshio Harada [the first author] on 28 November 2021).

**Table 9.** Opinion on the Kunugiyama observation site as the local landscape of the community.

| Landscape Like Hiraenoki | Landscape Not Like Hiraenoki | Can't Say Either Way |
| --- | --- | --- |
| 13 | 4 | 1 |

The reasons given for "feels like Hiraenoki" were "because I can see the autumn leaves of persimmons" (9 respondents), "because I can see the houses, roads, and other residential areas" (4 respondents), "because it is lush green" (1 respondent), and "because I have seen this view for a long time" (1 respondent). Opinions also included "It is sad that there are fewer houses" (1 respondent) and "It would be better if we could see Enoki area as well as Daira area" (1 respondent).

The reasons given for feeling "not Hiraenoki-like" were "We can only see Daira area and not Enoki area" (2 respondents), "The persimmon garden facing east in the morning sun is beautiful, but we can only see the south side from the observation site" (1 respondent), and "It is an unknown view from an unknown location" (1 respondent). One respondent chose "I can't say either way" because "I feel it's just an ordinary view of the countryside."

When asked how they felt about the construction site being visible from the observation site, the most common responses were "I want to improve it as a landscape" (5 respondents), "I have given up on it as inevitable" (5 respondents), and "I feel relieved that it is protected from disasters" (5 respondents).

Other responses were "I feel it will create awareness regarding this community being affected by the disaster" (4 respondents), "I feel that we suffered a major disaster" (3 respondents), "I feel anxious about the disaster" (1 respondent), "I do not want to see it if I could because it reminds me of the disaster" (1 respondent), "The landscape will fade if it turns black" (1 respondent), "I feel the construction is too slow" (1 respondent), "I feel it is just a matter of time because the greenery will return in 10 years" (1 respondent), and " I don't feel anything because I was not affected much by the disaster" (1 respondent).

3.2.10. Interaction Inside and Outside the Community after the Establishment of the Observation Site

As for whether the observation site had ever come up in conversation among the residents in the community, 10 respondents answered "yes" and 8 answered "no." However, 6 of the 10 who answered "yes" were HRC members, i.e., more than half of the residents who were not HRC members answered "no" (Table 10).

**Table 10.** Responses regarding whether the observation site had ever come up in conversation among community residents.

|  | **Yes** | **No** |
|---|---|---|
| HRC members | 6 | 1 |
| Non-HRC members | 4 | 7 |

With regard to whether interaction with the outside community had increased since the establishment of the observation site, 5 respondents answered "increased" and 13 answered "not changed" (Table 11). However, four of the five respondents who answered "increased" were HRC members. Only 1 of the 11 non-HRC residents responded that it had "increased". Therefore, there was a large difference between the perceptions of HRC members and those of non-HRC residents.

**Table 11.** Opinion on interaction with the outside community.

|  | **Increased** | **Not Changed** |
|---|---|---|
| HRC members | 4 | 3 |
| Non-HRC members | 1 | 10 |

In terms of opinions regarding outsiders coming in and out of the community, 16 responded "agree", 1 responded "disagree", and 1 responded "neither" (Table 12). One respondent who answered "no" stated that he was afraid of the COVID-19 virus being introduced from the outside. The one respondent who answered "neither" stated that he did not feel any particular concern regarding people from the outside coming in and out of the community, but that people had been passing noisily on the road in front of his house for some time, and he would be concerned if such people were to enter the community in the future.

**Table 12.** Opinion on the Kunugiyama observation site as local landscape of the community.

| **Agree** | **Disagree** | **Neither** |
|---|---|---|
| 16 | 1 | 1 |

## 4. Discussion

The observation project of the Hiraenoki Community revitalized local pride through landscape creation and involved numerous people outside the community, including the Asakura Extension Center, the Kyushu University Team, NPO Asa-Kuru staff, and children who participated in the Asa-Kuru activities. It provided an opportunity to form connections among people outside the community and residents inside the community. The Kunugiyama observation site is expected to become a "new symbol" for the community and may become an opportunity to recreate the community's identity.

Considering the roles of each entity involved in establishing the observation site, the HRC and cooperative residents played essential roles in planning, preparing project funds, building consensus within the community, and managing the observation site. Conversely, outsiders such as the Kyushu University Team and Asakura Extension Center served as advisors in their respective fields of expertise in the creation of the observation

site. They also acted as links between the Asa-Kuru NPO and the Hiraenoki Community. The NPO Asa-Kuru created contact points and communication opportunities between the community and the children. Furthermore, the HRC activities of the observation site led to new developments such as the formation of the "Hiraenoki Community Guard Group (*Hiraenoki Satomori Kai*)" a consultative body of the HRC, a citizens' group (NPO Asa-Kuru), and a university (Kyushu University Team). Therefore, the case of the Hiraenoki Community is regarded as an example as to how interaction with outsiders led to the reaffirmation of the community's appeal and dissemination to the outside community.

In contrast, our interviews revealed that the HRC faced several issues. Firstly, many residents found it challenging to maintain and manage the observation site. In addition, despite considerable participation in the work, awareness among HRC members and other residents in the community differed. For example, perception differed between men and women, among ex-Daira and ex-Enoki community residents, and in various age groups. The communication and sharing of ideas among them were insufficient. Furthermore, certain residents feel an "atmosphere of coercion" around the HRC activities, including the reconstruction tree-planting ceremony. This necessitates the sharing of objectives and issues between the HRC and other residents. Some residents also mentioned "activities targeting residents" as a challenge for the HRC. The HRC's philosophy of landscape creation is to "make the community a place where residents and others are happy to live". However, there is concern that collaboration with outsiders will obscure the original targets of the HRC activities. As in the case in Mashiki Town, which was damaged by the 2016 Kumamoto earthquake [9], the possibility that the landscape creation activities of the Hiraenoki Community can be a means of achieving revitalization of the local community depends on how much participation and understanding can be obtained from the residents in the maintenance activities for the Kunugiyama observation project.

The ex-residents who had moved out of the community were also insufficiently involved in promoting and sharing information regarding reconstruction activities and did not actively participate in the activities. Given the results of interviews with the ex-residents, it would be very difficult for some ex-residents to live in the Hiraenoki Community again. On the other hand, persimmon orchards and the invitation to the tree-planting ceremony from the HRC played roles in connecting ex-residents with the Hiraenoki Community. Therefore, how to utilize the Kunugiyama observation site as a "new communal place" of the Hiraenoki Community to maintain relationships among current residents and ex-residents is an important challenge. It is also essential to achieving the HRC's philosophy to "make the community a place where residents and others are happy to live".

In recent years, there has been increased attention on the "relationship population" in Japan. The Ministry of Internal Affairs and Communications of Japan describes the "relationship population" as a term that refers to people who are involved in a variety of ways with the community, i.e., neither the "settled population" who have moved to the area nor the "exchange population" who have come for sightseeing [19]. On the other hand, Sakuno argued that the "relationship population" should be regarded as one of a number of relationships between urban areas and agricultural and fishing village areas in this new era [20]. In communities facing the challenge of a shortage of residents who keep the community functioning due to population decline and aging, the "relationship population" is expected to become new bearers of the community. In our case study, some ex-residents still had a place attachment and strong network to the Hiraenoki Community even if it would be very difficult for them to live in the Hiraenoki Community again. Therefore, it appeared possible to retain them as a "relationship population" even if they could not be retained as a "settled population". Although our limited data does not allow for further discussion, it appears essential in the reconstruction process to find ways to maintain the "ex-settled population" as the "relationship population" in the post-disaster communities if some residents relocate out of communities after disasters. The reconstruction efforts through observation site creation in the Hiraenoki Community suggest the possibilities and challenges of resident-led landscape creation in the face of

aging, depopulation, and reconstruction through cooperation between reconstruction organizations inside and outside the community.

Finally, we were limited in that we could only obtain the opinions of one representative from each household. To understand the perceptions of a wide range of residents in the community, it is necessary to understand the views of other residents, including women. In addition, interviews with ex-residents who have relocated out of the communities after the NKTR are also needed for further discussion. It is also essential to continue participant observation and document resident-led reconstruction activities in the long term.

**Author Contributions:** Conceptualization, Y.H., A.I., K.A., N.S. and T.F.; methodology, Y.H., A.I., N.S. and T.F.; formal analysis, Y.H.; investigation, Y.H., A.I., N.S. and T.F.; writing—original draft preparation, Y.H.; writing—review and editing, T.O. and T.F.; supervision, K.A., N.S. and T.F.; project administration, N.S.; funding acquisition, N.S. All authors have read and agreed to the published version of the manuscript.

**Funding:** This work was supported by JSPS KAKENHI Grant Number JP18H04152.

**Institutional Review Board Statement:** All authors have taken a designated research ethics course required by the universities.

**Informed Consent Statement:** Informed consent was obtained from all subjects involved in the study.

**Data Availability Statement:** The data used to support the findings of this study are included within this article.

**Acknowledgments:** We are deeply grateful to the people of the Hiraenoki Community.

**Conflicts of Interest:** The authors declare no conflict of interest.

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
