# Peer review of "The Process and Challenges of Resident-Led Reconstruction in a Mountain Community Damaged by the Northern Kyushu Torrential Rain Disaster: A Case Study of the Hiraenoki Community, Asakura City, Fukuoka Prefecture, Japan"

_forests, doi:10.3390/f14040664_

Round 1
Reviewer 1 Report (Previous Reviewer 1)
Thank you for the many improvements to this paper. Adding the global context adds to the paper's relevance and general interest. The story of the inhabitants now feels more complete and poignant and motivates the reader to wish for good luck for their future.
24 In contrast, our investigation revealed => On the other hand, our investigation also revealed
27 communities facing depopulation and aging problems in tandem with => communities that are facing depopulation and aging problems, and are working with
69-70 Therefore, rehabilitating local landscapes in disaster-affected areas is not sufficient to recreate beautiful landscapes. => Rehabilitating local landscapes in disaster-affected areas is necessary but not sufficient to recreate enduring beautiful landscapes.
108 materialize => provide
109 most vulnerable => highly vulnerable
114 landscape carrier => landscape (" landscape carrier" is used almost exclusively by Chinese authors and may be a translation artifact – I do not know what it means)
377 were => are
Author Response
Please see the attachment.

Reviewer 2 Report (Previous Reviewer 2)
Thank you for your revisions. Although I am not sure you supply sufficient ethics information by international standards what you have will do.
Author Response
Please see the attachment.

This manuscript is a resubmission of an earlier submission. The following is a list of the peer review reports and author responses from that submission.
Round 1
Reviewer 1 Report
You have documented this situation very well. It is unavoidable that the numbers of people are very small and this is a single incident. On the other hand it seems likely that future studies will show that this story is typical for these situations. Much of the information will sound familiar to anyone who has been involved in community organizing. This story is relevant in many places outside Japan and to increase reader interest at the start it would be valuable to mention depopulation in Europe where it is also exacerbating the effects of climate change (such as fires in Spain).
It would be valuable for the reader to knw if community meetings were all in person o video meetings.
73 derive from them derive => derive from them
75 survive their villages => helping their villages survive
79 personal cutbacks => personnel cutbacks
124 is the Shiwa persimmon perhaps a variety rather than a separate species? Please identify with a citation
131 wherein addition => In addition
142 the meaning of abandoned in this context is unclear (perhaps remove)
160 does this sentence mean that local residents were asked to pay for road repair (which might create resentment)?
172 "Kunigiyama Observation" => the "Kunigiyama Observation" scenic overlook
223 initially => already
235 clean => clear
242 floor plan => detailed plan
261 floor plan => plan
323 it is worth mentioning if these species are all native species or if they include non-native ornamentals and whether there was consideration of the wildlife value for the restored vegetation
342 it is controversial and ambiguous to identify one member as the head of the household (and provokes the question were there no households with only one member?)
352 "I feel I forced to participate" => "I feel I was forced to participate"
389 unclear: target them
461 said => sad
476 comment about lonely, please review
Reviewer 2 Report
The paper presents a well covered area of community participation, although the context is a little more novel. However, it is disappointing that there is no linkage to the extensive literature that exists on community participation and decision-making and the difficulties it presents. As it is the paper is completely atheoretical in that sense, which is a shame is more might then be able to be taken out of a single case study.
The description is reasonable but i find a disjoint between the description and the conclusions. The latter emphasising the influence of outsiders but this, to me at least, is not strongly reflected in the body of the work. However, there should also be consideration of what the notions of outsider and local mean, as someone from outside or someone who has left a vilage may still have extremely strong networks and place attachment. But this point isn't really considered.
There also needs to be discussion of what ethics clearance and permissions were used for the study.